# Formation of Highly Efficient Perovskite Solar Cells by Applying Li-Doped CuSCN Hole Conductor and Interface Treatment

**DOI:** 10.3390/nano12223969

**Published:** 2022-11-10

**Authors:** In Seok Yang, You Jin Park, Yujin Hwang, Hoi Chang Yang, Jeongho Kim, Wan In Lee

**Affiliations:** Department of Chemistry and Chemical Engineering, Inha University, Incheon 22212, Korea

**Keywords:** perovskite solar cell, inorganic hole-transporting material (HTM), CuSCN, Li-doped CuSCN, interface control, PCPDTBT

## Abstract

Li-doped CuSCN films of various compositions were applied as hole-transporting material (HTM) for mesoscopic perovskite solar cells (PSCs). Those films of ~60 nm thickness, spin-coated on the perovskite layer, exhibit significantly higher crystallinity and hole mobility compared with the pristine CuSCN films. Among them, 0.33% Li-doped CuSCN (Li0.33:CuSCN) shows the best performance as the HTM of mesoscopic PSC. Furthermore, by depositing a slight amount of PCPDTBT over the Li0.33:CuSCN layer, the *V_OC_* was increased to 1.075 V, resulting in an average PCE of 20.24% and 20.65% for the champion device. These PCE and *V_OC_* values are comparable to those of PSC using spiro-OMETAD (PCE: 20.61%, *V_OC_*: 1.089 V). Such a remarkable increase can be attributed to the penetration of the PCPDTBT polymer into the grain boundaries of the Li0.33:CuSCN film, and to the interface with the perovskite layer, leading to the removal of defects on the perovskite surface by paving the non-contacting parts, as well as to the tight interconnection of the Li0.33:CuSCN grains. The PSC device with Li0.33:CuSCN showed a high long-term stability similar to that with bare CuSCN, and the introduction of PCPDTBT onto the perovskite/Li0.33:CuSCN further improved device stability, exhibiting 94% of the initial PCE after 100 days.

## 1. Introduction

Ever since alkylamine lead halide perovskite materials were used as light absorbers of solar cell for the first time in 2009 [1] and their solid-state device was developed in 2012 [2], remarkable progress has been accomplished in improving the power-conversion efficiency (PCE) of perovskite solar cells (PSCs). Currently, a certified PCE of over 25%, comparable to that of a single-crystal Si solar cell, has been achieved [3], and diverse studies are in progress for the commercial application of PSC devices [4,5,6,7,8]. In typical PSC architecture, the perovskite light-absorbing layer is sandwiched between an electron-transporting layer (ETL) and a hole-transporting layer (HTL) to collect photogenerated charges efficiently. Comparatively higher PCEs are achieved from the NIP-type-device architecture employing TiO_2_ or SnO_2_ layers as the ETL, and organic hole-conductors such as spiro-OMETAD, PTAA and P3HT as the HTL [9,10,11,12]. 

As an alternative to organic hole-transporting materials (HTMs), inorganic HTMs have also attracted much attention because of low material cost and high stability, which are essential requirements for the commercialization of PSC devices. Furthermore, inorganic HTMs exhibit many outstanding properties based on their high hole-mobility, suitable HOMO level with facile tunability, and large band-gap that is non-resonant with visible light. However, it is difficult to fabricate high-quality films of inorganic HTMs by a simple solution process, and hence it is challenging to form a defect-free interface between the perovskite and inorganic HTM layers. Thus far, various inorganic materials such as CuI [13,14], CuSCN [15,16,17], NiO_x_ [18,19,20,21,22], CuGaO_2_ [23,24], CuCrO_2_ [25,26], CuS [27], CuO_x_ [28,29], and others [30,31,32,33,34], have been applied as HTMs. Among them, we believe that one of the most promising materials is CuSCN, because its organometallic structure facilitates the fabrication of uniform films by solution process at low temperature.

Conventionally, a doctor blade method has been used for the coating of CuSCN films [15,16,35]. However, PCEs of only 12–13% were achieved for the mesoscopic PSCs fabricated by this method, probably owing to the damage of the underlying perovskite layer by the alkylsulfides used for the coating solution [35]. More recently, a spin coating process has been successfully developed to alleviate damages to the perovskite layer [36,37,38,39,40], and a high PCE of over 20% was reported [38]. In addition, Lee et al. reported a novel spray-deposition method, which is considered to be very useful for large-area deposition [41]. In spite of various efforts to develop deposition methods for CuSCN films [17,35,36,37,38,39,40,41,42,43,44,45,46], however, the PSC devices employing CuSCN still exhibit a significantly lower photovoltaic (PV) performance than those with organic HTMs.

It has been reported that the poor interface between the perovskite and CuSCN layers is responsible for the relatively lower PV properties of the CuSCN-based PSC devices [35,39,47,48,49,50,51,52]. In particular, during the deposition of the CuSCN layer, the surface of the underlying perovskite layer can be damaged, thus inducing the formation of defects on its surface. Moreover, there is an unavoidable contact problem between the CuSCN and the perovskite layers. As the coated CuSCN film is crystallized to form polycrystalline structures, from a microscopic point of view some parts of the coated CuSCN layer do not make contact with the perovskite layer, although major parts contact successfully [35]. At the non-contacting part of the perovskite/CuSCN interface, photogenerated holes can be accumulated without transportation to the CuSCN layer. Consequently, charge recombination can take place at this site, leading to the decrease of *V_OC_*.

Recently, interface control by introducing functional molecules at the perovskite/CuSCN interface has been attempted, to passivate the perovskite surface and to interconnect the perovskite and the CuSCN layer [48]. Such interface control has allowed for the achievement of significantly higher *V_OC_* and improved long-term stability. In addition, as a means of minimizing the damage to the perovskite layer during the coating of the CuSCN layer, a thin two-dimensional (2D) layer of perovskite was introduced as a supplementary surface over the main 3D-perovskite layer [39,40]. The presence of the 2D-perovskite layer protects the 3D-perovskite layer from the damage, leading to significant improvement in device stability as well as in PV performance. 

The hole mobility of CuSCN has been reported to be in the range of 0.01–0.1 cm^2^V^−1^s^−1^ [53,54,55]. Although the hole mobility of CuSCN is much higher than that of organic HTMs, it is relatively lower than those of several inorganic HTMs such as several delafossite oxides, CuI, Cu_2_O and others [54,55,56]. Since it is desirable to have a higher hole mobility for efficient charge collection and the achievement of high PV performance, we attempted to enhance the hole mobility by doping the Li ion into the Cu-site in the hexagonal CuSCN crystal structure. Previously, there have been attempts to dope SCN^−^ into CuSCN for application to solid-state dye-sensitized solar cells [57], and a recent report indicated that Cl_2_-doped CuSCN was effective in improving hole mobility, PV properties, and the stability of PSC devices [58]. 

It has been known that the hole mobility of CuSCN originates from the intrinsic vacancies of Cu^+^ ions occupying the tetrahedral sites of the hexagonal β-CuSCN structure [59,60,61]. The energy states near the valence band maximum (VBM) of β-CuSCN are mainly contributed by Cu 3d with partial hybridization from the S 3p orbital. Therefore, Cu vacancies can serve as the acceptor states over the VBM of CuSCN, and are responsible for the characteristic hole mobility [61]. For the Li-doped CuSCN, the doping of Li^+^ into the Cu^+^ site will also provide acceptor states over the VBM of CuSCN, because Li^+^ has empty 3d atomic orbitals. Furthermore, the density of acceptor states of CuSCN can be controlled by varying the doping concentration of Li^+^. 

Here we found that Li-doped CuSCN (Li:CuSCN) has a higher hole mobility by an order of magnitude than CuSCN, whereas the coated Li:CuSCN film exhibits larger grains with higher crystallinity that might degrade the perovskite/Li:CuSCN interface. For additional interface control, we introduced poly [2,6-(4,4-bis-(2-ethylhexyl)-4H-cyclopenta [2,1-b;3,4-b′]dithiophene)-alt-4,7(2,1,3-benzothiadiazole)] (PCPDTBT), which contains many sulfur atoms that can bind well to both the perovskite [62] and the copper thiocyanate [48]. In particular, by introducing a slight amount of PCPDTBT to the perovskite/Li:CuSCN, a PCE of over 20% was achieved, and the observed *V_OC_* and PCE values were comparable to those of PSC devices employing spiro-OMETAD. Moreover, we characterized the Li:CuSCN structures with various doping concentrations and the properties of their films, and systematically analyzed the performances of PSC devices of various compositions, in order to understand the role of PCPDTBT in enhancing the PV properties of PSC devices.

## 2. Materials and Methods

### 2.1. Preparation of Li-Doped CuSCN

The Li-doped CuSCN of various compositions were prepared by reacting stoichiometric amounts of CuSO_4_∙5H_2_O, LiCl and KSCN in an aqueous solution, by modifying the procedure of bare CuSCN preparation reported previously [63]. To obtain 0.33 mol% Li-doped CuSCN (Li_0.0033_Cu_0.9967_SCN or Li0.33:CuSCN), 4 mol% LiCl was introduced, as most of the added LiCl was washed out during the synthesis process. In addition, Li0.18:CuSCN and Li0.69:CuSCN were prepared by adding 2 and 10 mol% LiCl, respectively. In order to prepare the Li0.33:CuSCN, 236.8 mg CuSO_4_∙5H_2_O and 1.69 mg LiCl were dissolved in 60 mL distilled water in a beaker, while 194.4 mg KSCN, which was used as the source of SCN^−^, was dissolved in 40 mL water in another beaker. The two solutions (with a molar ratio of Cu + Li to SCN = 1:2) were then mixed, and stirred vigorously at ambient condition. Initially, a blackish precipitate identified as Li_0.0033_Cu_0.9967_(SCN)_2_ was formed, and then it was gradually converted to a pale purple-colored powder, by washing several times with distilled water. During the washing process, excess SCN^−^ was removed to form Li0.33:CuSCN. A pale purple-colored Li0.33:CuSCN powder was then collected, and dried in a vacuum oven at 100 °C for 12 h.

### 2.2. Fabrication of FTO/TiO_2_/Perovskite

To prepare a TiO_2_ blocking layer, an approximately 20 nm-thick Ti layer was deposited on the FTO glass (Pilkington, TEC-8, Lancashire, UK), using an RF magnetron sputtering system (A-Tech system, Incheon, Korea), followed by oxidation at 500 °C for 30 min in air. Then, a mesoporous TiO_2_ layer with ~180 nm thickness was prepared by the spin-coating of TiO_2_ paste derived from the 50 nm-sized TiO_2_ nanoparticles (NPs), at 6000 rpm for 20 s and subsequent calcination at 500 °C for 30 min in air [64]. A double cation perovskite with the chemical formula of FA_0.9_MA_0.1_Pb(I_0.9_Br_0.1_)_3_ (FA: formamidinium, MA: methylammonium) was applied as a perovskite light-absorber, and its film was prepared by a one-step process employing the coating solution described below. That is, 845 mg FAPbI_3_ (ShareChem, Daejeon, Korea) and 65 mg MAPbBr_3_ (ShareChem) were dissolved in the mixture of 880 μL dimethylformamide (DMF, 99.8%, Sigma Aldrich, St. Louis, MO, USA) and 120 μL dimethyl sulfoxide (DMSO, 99.9%, Sigma Aldrich). Additionally, 33 mg methylammonium chloride (Sigma Aldrich) was added to this solution to improve the stability of the perovskite phase. A total of 40 μL of the prepared coating solution was dropped onto the FTO/TiO_2_ substrate, and then spun at 4000 rpm for 20 s, while 1 mL of diethyl ether was dropped onto the substrate after 10 s of spinning had elapsed. The coated film was baked at 65 °C for 1 min and then at 150 °C for 10 min, on a hot plate.

### 2.3. Preparation of HTM Layer and Metal Contact

The CuSCN or various Li:CuSCN layers were coated on the FTO/TiO_2_/FA_0.9_MA_0.1_Pb(I_0.9_Br_0.1_)_3_ substrate by a modified spin-coating technique, employing a fast-evaporation method [65,66]. A total of 50 μL of 0.15 M CuSCN (or Li:CuSCN) solution in diethyl sulfide was dropped onto the substrate, followed by immediate spinning at 2000 rpm for 20 s. After 2 s of spinning, hot wind was blown onto the substrate, to minimize the damage to the perovskite layer by the solvent. After spin-coating, the substrate was heat-treated on a hot plate at 85 °C for 3 min, to evaporate the residual diethyl sulfide and to induce the crystallization of CuSCN (or Li:CuSCN). Optionally, for the PCPDTBT (Aldrich, molecular weight: 7000–20,000) treatment, a very dilute PCPDTBT solution in chlorobenzene (0.2 mg/mL) was dropped onto the FTO/TiO_2_/FA_0.9_MA_0.1_Pb(I_0.9_Br_0.1_)_3_/Li:CuSCN substrates, followed by spinning at 5000 rpm for 30 s. As a control experiment, spiro-OMeTAD was used as the HTM. That is, 72.3 mg spiro-OMeTAD was dissolved in 1 mL chlorobenzene containing 28.8 μL 4-tertbutylpyridine and 17.5 μL lithium bis(trifluoromethylsulphonyl) imide (520 mg/mL in acetonitrile). The as-prepared 50 μL spiro-OMeTAD solution was dropped onto the FTO/TiO_2_/FA_0.9_MA_0.1_Pb(I_0.9_Br_0.1_)_3_ substrate, followed by spin-coating at 4000 rpm for 30 s. As a metal contact, a Au layer of ~60 nm thickness was deposited over the HTM layer, using a thermal evaporator (Korea Vacuum Tech., Gimpo, Korea).

### 2.4. Measurements and Characterizations

Photocurrent density-voltage (*J-V*) curves of PSC devices were measured at ambient condition under an irradiation of AM 1.5 G one sun light. The PSC devices were masked with a nonreflective black metal aperture to define the active area, typically 0.122 cm^2^, which was measured using an optical microscope. For measurement of *J-V* curves, the applied voltages were scanned in the reverse direction, with a scan rate of 200 mV s^−1^, and the dwelling time before the voltage scan was 50 ms. Incident photon-to-current efficiency (IPCE) spectra were obtained in the wavelength range of 350–850 nm, using an IPCE measurement system (PV Measurements, Inc., Boulder, CO, USA). The hole mobilities of the HTM samples were evaluated simply, using a Hall effect measurement system (HMS-3000, Ecopia, Inc., Toronto, ON, Canada). For the measurement, each HTM was coated on a Pyrex glass with a thickness of ~300 nm.

The time-resolved photoluminescence (TR-PL) was measured with a time-correlated single photon counting (TCSPC) spectrometer (FluoTime 200, PicoQuant, Berlin, Germany). The excitation wavelength and the detection wavelength of the TR-PL measurement were 393 nm and 768 nm, respectively, and its time resolution was ~200 ps. The transient absorption spectroscopy (TAS) measurement was performed with a nanosecond transient absorption spectrometer (LP980, Edinburgh Instruments, Livingston, UK). The pump laser pulses of 355 nm wavelength, 5 ns duration, and 0.34 mJ per pulse were focused to a spot of 10 × 18 mm^2^ from the FTO substrate side of the FTO/perovskite/HTM sample at ambient condition, giving the excitation fluence of 0.19 mJ/cm^2^ per pulse. A pulsed (6 ms duration) Xe arc lamp was used as a probe source. The transient change in the absorption of the sample was measured in real time with a PMT detector and a digital oscilloscope of 200 MHz bandwidth. The time resolution of the TA measurement was 7 ns. The TAS measurements were repeated multiple (at least three) times for each type of PSC sample, to check the reproducibility of the TA data.

The elemental composition of the Li-doped CuSCN samples was analyzed by inductively coupled plasma optical emission spectroscopy (ICP-OES) (OPTIMA 8300, Perkin Elmer, Waltham, MA, USA) using CuSCN, LiSCN, and their mixtures, as standards.

## 3. Results and Discussion

### 3.1. Characterization of Li-Doped CuSCN Powders and Films

Li-doped CuSCN with various compositions (Li_x_Cu_1−x_SCN, 0 ≤ x ≤ 0.0069) was prepared from the reaction of CuSO_4_∙5H_2_O, KSCN and a stoichiometric amount of LiCl. The radius of the Li^+^ ion (73 pm) is nearly identical to that of the Cu^+^ ion (74 pm), which occupies the tetrahedral site of the hexagonal β-CuSCN structure. Li^+^ has the highest polarizability among the alkali metal ions, although Li^+^ is not as soft as Cu^+^ in regard to cationic softness [67]. In this regard, Li^+^ is considered one of the suitable cations that can replace Cu^+^ in the β-CuSCN structure. The XRD patterns of Li:CuSCN powders with various Li compositions are shown in Figure 1a, indicating that all of the prepared Li-doped CuSCN samples are in the pure β-CuSCN phase, without an impurity phase within the detection limit. In addition, the doping of Li^+^ leads to relatively sharper XRD peaks, implying that the incorporation of Li^+^ ion induces higher crystallinity.

An energy dispersive X-ray (EDX) analysis was performed on the bare CuSCN, Li0.33:CuSCN, and Li0.69:CuSCN (0.69 mol% Li-doped CuSCN). Due to the low elemental signal, which is characteristic of lithium, Li^+^ in the Li-doped CuSCN samples was not identified, as shown in Appendix A. In addition, the K^+^ ion was not detected at all for the as-prepared Li-doped CuSCN, suggesting that KSCN used as the source of SCN^−^ does not remain in the Li:CuSCN samples.

The elemental composition of the Li-doped CuSCN samples was analyzed using ICP-OES. The nominal compositions of the Li-source in the reactant were 2, 4, and 10 mol% relative to Li + Cu, but the actual Li composition in the prepared Li_x_Cu_1−x_SCN was determined to be 0.18, 0.33 and 0.69 mol%, respectively. Much lower compositions of Li in the products suggests that most of the Li-source was washed out during the synthesis process. Therefore, more reliable synthetic methods for doping Li into CuSCN have to be developed.

The hole mobility and conductivity of bare CuSCN and Li-doped CuSCN of various compositions are listed in Appendix A. It was determined that 0.33 mol% doping of Li^+^ induces the highest hole mobility of 1.42 cm^2^V^−1^s^−1^, which is approximately one order of magnitude higher than that of the pristine CuSCN (0.15 cm^2^V^−1^s^−1^).

The valence band maximum (VBM) of the bare CuSCN and Li0.33:CuSCN was determined by ultraviolet photoelectron spectroscopy (UPS). As shown in Appendix A, E_VBM_ of bare CuSCN was determined to be 5.67 eV from the cutoff energy (E_cutoff_) of 16.28 eV and E_VBM_—E_F_ value of 0.73 eV, with the relationship of E_F_ = 21.22 eV (He I)—E_cutoff_. In the same way, E_VBM_ of Li0.33:CuSCN was determined to be 5.70 eV. According to the UPS analysis, the VBM level was not appreciably changed by the doping of Li^+^, and the measured VBM values agreed with the ones reported in previous studies [50,53,59,60,61]. The VBM values were considerably more positive than the well-known HOMO level of 5.3 eV for the pristine CuSCN. However, as previously reported, the discrepancy between these two values was caused by the presence of band-tail states above the VBM of CuSCN [68].

Plan-view and cross-sectional-view SEM images of the bare CuSCN and Li0.33:CuSCN films coated over the perovskite layer are illustrated in Figure 2. The thickness of those films coated by the spin coating method was determined to be approximately 60 nm. Compared with the bare CuSCN film, the Li0.33:CuSCN film shows a more crystallized grain structure. That is, comparing the SEM images of the grains in the insets of Figure 2b,c, the grains in the Li0.33:CuSCN film are larger than those in the bare CuSCN. In addition, the XRD patterns (Figure 1b) for the coated films over the glass/perovskite reveal that the Li0.33:CuSCN film shows sharper diffraction peaks with higher intensities than the bare CuSCN film, suggesting that doping of Li^+^ ion increases the crystallinity of the CuSCN film.

### 3.2. PV Properties of PSCs with Li-Doped CuSCN HTL

The Li-doped CuSCN films of various compositions were applied as HTL of the mesoscopic perovskite solar cells. The *J-V* curves of mesoscopic PSC devices employing MA_0.1_FA_0.9_Pb(I_0.9_Br_0.1_)_3_ as a perovskite light-absorber, are shown in Figure 3, and their PV parameters are listed in Table 1. Li:CuSCN films doped with 0.18 and 0.33% Li^+^ provide appreciably improved PV performances, compared with the pristine CuSCN film, whereas 0.69% doping deteriorates the cell performance (Figure 3a). Among the films, the Li0.33:CuSCN film showed the optimum performance, but the improvement of PV properties was not significant. The PSC device with Li0.33:CuSCN (PSC-Li0.33:CuSCN) exhibits an average PCE of 19.06% with *V_OC_* of 1032 mV, *J_SC_* of 24.40 mA cm^−2^, and *FF* of 75.69%, whereas the device with bare CuSCN (PSC-CuSCN) exhibits an average PCE of 18.03%, with *V_OC_* of 1018 mV, *J_SC_* of 23.86 mA cm^−2^, and *FF* of 74.24%.

Although the Li0.33:CuSCN possesses higher hole mobility than the bare CuSCN, the increase of *V_OC_* and PCE was not significant. We believe that this is closely related to the larger grain size of the Li0.33:CuSCN film formed over the perovskite layer. Its larger grains deteriorate the interconnection among the individual Li0.33:CuSCN grains and the contact with the perovskite surface. As a result, it is likely that more defects are formed at the perovskite/Li0.33:CuSCN interface, leading to an increase in charge recombination at the interface. Thus, these two opposite effects cancel each other out, resulting in no significant improvement of *V_OC_* and PCE in the PSC-Li0.33:CuSCN device.

### 3.3. Improvement of PV Properties by Introducing PCPDTBT

As a means of improving the interconnection among the Li0.33:CuSCN grains and the inter-layer contacts at the perovskite/Li0.33:CuSCN interface, we coated a very small amount of PCPDTBT over the Li0.33:CuSCN layer, using the procedure illustrated in Figure 4. Specifically, a very dilute PCPDTBT solution in chlorobenzene of 0.2 mg/mL concentration, which is only ~1% of the concentration typically used for the coating of organic HTMs, was spin-coated over the Li0.33:CuSCN layer. PCPDTBT polymers (see Figure 4) contain numerous sulfur atoms which have been known to have a strong binding affinity to the Cu atom in the CuSCN [48] as well as the Pb in the perovskite [59]. Moreover, the HOMO level of PCPDTBT (5.3 eV) is located at the same energy level as that of the CuSCN. Thus, the presence of PCPDTBT at the grain boundaries of the Li0.33:CuSCN layer and the perovskite/Li0.33:CuSCN interface would be greatly advantageous for the interconnections of the Li0.33:CuSCN grains and the contact between perovskite and Li0.33:CuSCN.

Figure 2d,g,h show plan-view and cross-sectional-view SEM images of PCPDTBT-coated Li0.33:CuSCN (Li0.33:CuSCN/PCPDTBT) films. The plan-view image in Figure 2d shows that the porous structure of the Li0.33:CuSCN film is partially paved by PCPDTBT. In addition, the cross-sectional image in Figure 2g exhibits the fact that the interface of the perovskite/Li0.33:CuSCN was significantly changed, although the PCPDTBT was coated simply on top of the Li0.33:CuSCN layer. Noticeably, as seen in the magnified image (Figure 2h), a new substance is present at the interface. We speculate that the material present in the dashed ellipse in Figure 2h is the PCPDTBT. It is inferred that some of the coated PCPDTBT penetrated the porous Li0.33:CuSCN layer to reach the perovskite/Li0.33:CuSCN interface. Figure 2j–l shows atomic force microscope (AFM) images of bare perovskite, perovskite/Li0.33:CuSCN, and perovskite/Li0.33:CuSCN/PCPDTBT surfaces. By coating PCPDTBT on the perovskite/Li0.33:CuSCN, the average roughness (R_av_) of Li0.33:CuSCN decreased only slightly, suggesting that the amount of PCPDTBT present on the surface of the Li0.33:CuSCN layer is very small.

XPS depth profiles were obtained to analyze the presence of PCPDTBT throughout the perovskite/Li0.33:CuSCN/PCPDTBT multilayered films. As shown in Appendix A, the elemental compositions of Cu, Pb, and C were monitored as a function of etch time. The presence of PCPDTBT can be identified from the distribution of element C. Broadly speaking, the Li0.33:CuSCN layer is present in the 0−250 s etch-time range, and the perovskite layer appears in the etch-time range higher than 250 s. The C content in the perovskite/Li0.33:CuSCN/PCPDTBT film was significantly higher in the 0−300 s range compared with that in the perovskite/Li0.33:CuSCN. This suggests that PCPDTBT is present not only on the top of the Li0.33:CuSCN layer, but also inside the Li0.33:CuSCN layer and at the interface of the perovskite layer, which is consistent with the observations from the SEM images in Figure 2.

Consequently, a significant amount of PCPDTBT permeates into the Li0.33:CuSCN grain boundaries (or pinholes), and induces a tight interconnection of Li0.33:CuSCN grains, because the sulfur atoms of the PCPDTBT can have strong interactions with the Cu atom of Li0.33:CuSCN. In addition, the PCPDTBT can penetrate down to the interface of the perovskite/Li0.33:CuSCN. Therefore, the non-contacting part at the interface can be paved by the PCPDTBT, resulting in the removal of the defects on the perovskite surface by the passivation of the Pb atoms.

Figure 3b shows the *J-V* curves of PSC-Spiro, PSC-CuSCN, PSC-Li0.33:CuSCN, PSC-CuSCN/PCPDTBT, and PSC-Li0.33:CuSCN/PCPDTBT. When the PCPDTBT was deposited over the bare CuSCN layer, the *V_OC_* and PCE of the devices increased only a little, to 1.048 V and 19.12%, respectively. We believe that such limited improvement of PV properties with the treatment with PCPDTBT must be closely related to the low crystallinity of CuSCN grains, exhibiting low porosity, as seen in the SEM images in Figure 2b,e. As a result, as shown in Figure 2i, the perovskite/CuSCN interface was not altered appreciably by PCPDTBT coating, suggesting that PCPDTBT did not permeate through the CuSCN layer. In contrast, when the PCPDTBT was coated over the Li0.33:CuSCN layer, a remarkable enhancement of PV performance was achieved. In particular, *V_OC_* was increased to 1.075 V, leading to an average PCE of 20.24%, although *J_SC_* was not appreciably changed. It is noteworthy that the PSC device employing the Li-doped CuSCN and treated with PCPDTBT exhibited *V_OC_* and PCE values comparable to the PSC-Spiro, which shows *V_OC_* of 1.089 V and PCE of 20.61%.

In addition, we performed several control experiments to understand the role of PCPDTBT in enhancing PV properties. First, without CuSCN, we applied a pristine PCPDTBT as the HTM, by spin-coating with a solution of 0.2 mg/mL concentration; the achieved PCE was determined to be only 9.40%, as shown in Appendix A and Table 1. Second, when the PCPDTBT was applied as the HTM but coated with a solution of 20 mg/mL concentration, which is a typical concentration for coating organic HTMs, the fabricated device exhibited a PCE of only 8.52% (Appendix A and Table 1). These results suggest that the pristine PCPDTBT itself is not an efficient HTM. In other words, the enhanced PCE of PSC-Li0.33:CuSCN/PCPDTBT does not originate from the hole-transport property of PCPDTBT but from its unique roles of interconnecting the individual Li0.33:CuSCN grains and effectively improving the contacts at the perovskite/Li0.33:CuSCN interface, as described in Figure 4b. Third, we coated the PCPDTBT with a 0.2 mg/mL solution over the perovskite layer before the deposition of Li0.33:CuSCN. Although the fabricated PSC-PCPDTBT/Li0.33:CuSCN device showed a slightly improved PCE (19.27%) compared with PSC-Li0.33:CuSCN (Appendix A and Table 1), its PCE was significantly lower than that of PSC-Li0.33:CuSCN/PCPDTBT. Thus, it is indicated that a coating of PCPDTBT on the Li0.33:CuSCN layer is a more effective way to improve the perovskite/Li0.33:CuSCN interface. This is because the damage to the perovskite surface can be reduced, and the grain boundaries (or nanopores) formed in the Li0.33:CuSCN layer can be successfully paved, through this process.

IPCE spectra were obtained for several PSC devices, as shown in Figure 3c. The *J_SC_* values acquired from the integration of the IPCE spectra are close to the values acquired from the *J-V* curves in Figure 3b. *J-V* curves for forward and backward scans were acquired from PSC-Li0.33:CuSCN/PCPDTBT. As shown in Appendix A, PSC-Li0.33:CuSCN exhibits a significantly larger hysteresis than the PSC-Spiro. However, PSC-Li0.33:CuSCN/PCPDTBT exhibits very little hysteresis in the *J-V* measurement, as seen in Figure 5a. In fact, with the PCEs of 20.24% and 19.82% from the reverse and forward scans, respectively, its hysteresis is comparable to that of the PSC-Spiro. Such significant reduction of hysteresis in PSC-Li0.33:CuSCN/PCPDTBT can be attributed to the presence of PCPDTBT at the perovskite/Li0.33:CuSCN interface. That is, most of the defects formed at the perovskite/Li0.33:CuSCN interface were removed by PCPDTBT treatment, thereby removing charge accumulation at the interface.

For the PSC-Li0.33:CuSCN treated with PCPDTBT, a steady-state photocurrent and power output at the maximum power point were monitored, as shown in Figure 5b. The power output was stable, and the acquired PCE of 19.89% is close to the average value obtained from the forward and backward scans. Furthermore, this process provides a highly reproducible PV performance. As illustrated in Figure 5c, the PSC-Li0.33:CuSCN/PCPDTBT devices show much narrower distributions of PCE values, compared with PSC-CuSCN or PSC-Li0.33:CuSCN. The PCE of the champion device was 20.65% (Figure 5d), which is one of the highest PCE values reported for the CuSCN-based PSC devices thus far.

We monitored the device stability under ambient condition for the as-prepared PSC-Spiro, PSC-CuSCN, PSC-Li0.33:CuSCN and PSC-Li0.33:CuSCN/PCPDTBT. Specifically, the PV properties of those PSC devices stored at ambient condition with a relative humidity of 25 ± 5% were monitored for up to 100 days. As shown in Figure 5e, the bare PSC-CuSCN shows significantly higher long-term stability than the PSC-Spiro as observed by other reports [37,38,39,41,48]. In addition, PSC-Li0.33:CuSCN exhibits similar stability as PSC-CuSCN. After 100 days, the PSC-CuSCN and PSC-Li0.33:CuSCN exhibit 87% and 88%, respectively, of their own initial PCE. This suggests that a very small amount of Li doping into the CuSCN does not affect the device stability. We also monitored the effect of the PCPDTBT treatment on the device stability. Remarkably, PSC-Li0.33:CuSCN/PCPDTBT showed further improved long-term stability. After 100 days, 94% of the initial PCE was maintained for the PSC-Li0.33:CuSCN treated with PCPDTBT. We infer that the PCPDTBT treatment removes defect sites and blocks ion migration at the perovskite/Li0.33:CuSCN interface, resulting in the higher stability of the PSC device.

### 3.4. Charge Injection Dynamics

To examine the dynamics of charge injection to HTM, we measured time-resolved photoluminescence (TR-PL) for the PSC devices investigated in this work, following the protocol described in previous reports [41,48]. As shown in Figure 6a and Table 2, it can be seen that the TR-PL decay becomes faster in the order of bare Perovskite (Perov) < Perov/CuSCN < Perov/Li0.33:CuSCN < Perov/Li0.33:CuSCN/PCPDTBT. In general, as the photogenerated holes are injected into the HTM more efficiently, the PL decay becomes faster. The result of the TR-PL measurement indicates that Li0.33:CuSCN is a more efficient hole acceptor than pristine CuSCN, and that the treatment with PCPDTBT improves the hole transport even further.

We also performed a transient absorption spectroscopy (TAS) measurement, to examine the dynamics of charge injection to the HTM in the PSC devices. The temporal decays of the TA spectra at 780 nm are shown in Figure 6b, and the lifetimes of the decays determined from their single-exponential fits are listed in Table 2. It can be seen that the initial amplitude of the TA signal at t = 0 becomes smaller (that is, less negative) in the order of bare Perov > Perov/CuSCN ≈ Perov/Li0.33:CuSCN/PCPDTBT > Perov/Li0.33:CuSCN. The smaller TA signal at t = 0 means that the TA signal decays faster within the time resolution of our TA measurement (7 ns). In addition, it can be seen in Table 2 that the lifetime of the TA decay at 780 nm becomes shorter in the same order as the initial amplitude of the TA signal at t = 0. The shorter lifetime of the TA signal indicates either more efficient hole injection to the HTM or more active charge recombination after hole transfer. These processes give opposite effects on the performance of solar cells, with the hole transfer improving the cell performance and the other deteriorating the cell performance. In fact, a PSC device that exhibits faster TA decay gives higher *V_OC_* and PCE values, as can be seen in Table 1, suggesting that the decay of the TA signal at 780 nm reflects the dynamics of hole injection more dominantly, rather than that of charge recombination. The result of the TAS measurement implies that the hole injection to Li0.33:CuSCN occurs more efficiently than to pristine CuSCN, which agrees with the TR-PL result, although the effect of the PCPDTBT treatment was not well reflected in the TA signal.

## 4. Conclusions

Li0.33:CuSCN demonstrates a notably high hole mobility (1.42 cm^2^V^−1^s^−1^), which is approximately one order of magnitude higher than that of the bare CuSCN. Accordingly, the PSC device employing Li0.33:CuSCN as the HTM exhibits higher *J_SC_* and *FF*, but its *V_OC_* increase is only 14 mV, as the Li0.33:CuSCN film with large grains cause the interface with the perovskite layer to deteriorate. However, when a small amount of PCPDTBT was deposited over the Li0.33:CuSCN layer, the *V_OC_* was increased from 1.032 V to 1.075 V and the PCE was increased to 20.24% on average, and 20.65% for the champion device. Consequently, the achieved PCE and *V_OC_* values were comparable to those of the PSC employing spiro-OMETAD (PCE: 20.61%, *V_OC_*: 1.089 V). Such a remarkable increase in PCE and *V_OC_* can be attributed to the penetration of polymeric PCPDTBT into the grain boundaries of Li:CuSCN and the perovskite/Li0.33:CuSCN interface. That is, the treatment with a very small amount of PCPDTBT effectively interconnects the Li0.33:CuSCN grains, and paves the non-contacting parts formed at the interface. The PSC device with Li0.33:CuSCN shows similar long-term stability to the device with bare CuSCN, suggesting that Li^+^ ions occupying the Cu^+^ sites are not mobile. The introduction of PCPDTBT to the pervskite/Li0.33:CuSCN interface further improves stability, with 94% of the initial PCE being maintained after 100 days.

## Figures and Tables

**Figure 1 nanomaterials-12-03969-f001:**
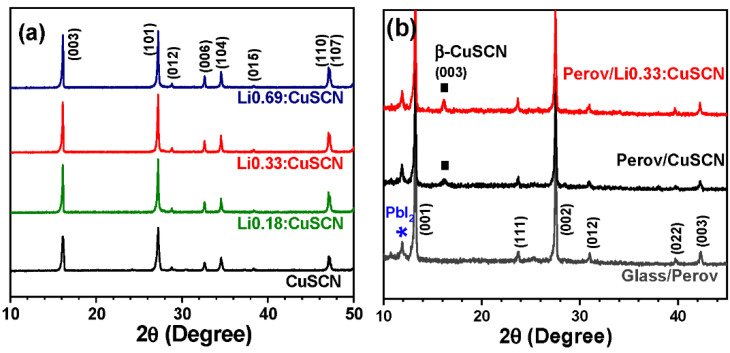
XRD patterns of the as-prepared CuSCN and Li:CuSCN powders with various Li^+^ doping concentrations (**a**), and the CuSCN and Li0.33:CuSCN films coated over the FA_0.9_MA_0.1_Pb(I_0.9_Br_0.1_)_3_ perovskite layer (**b**).

**Figure 2 nanomaterials-12-03969-f002:**
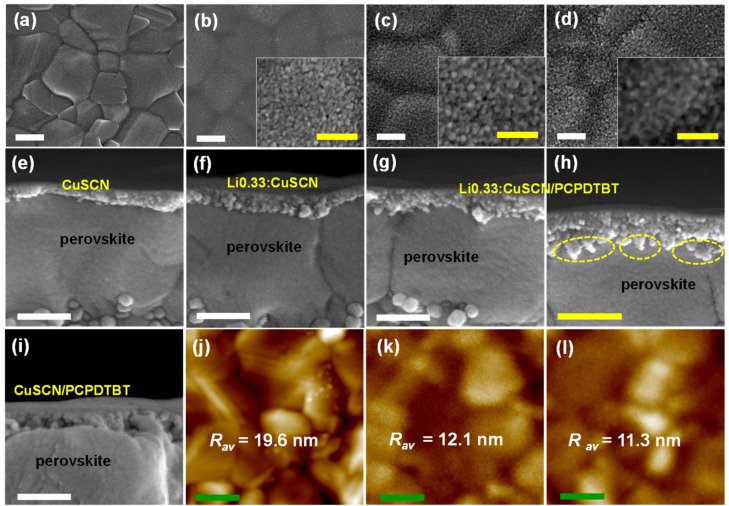
Plan-view SEM images of the bare perovskite (**a**), perovskite/CuSCN (**b**), perovskite/Li0.33:CuSCN (**c**), and perovskite/Li0.33:CuSCN/PCPDTBT (**d**). Insets in b, c, and d are magnified images of the corresponding figures. Cross-sectional SEM images of perovskite/CuSCN (**e**), perovskite/Li0.33:CuSCN (**f**), perovskite/Li0.33:CuSCN/PCPDTBT (**g**,**h**), and perovskite/CuSCN/PCPDTBT (**i**). AFM images of the bare perovskite (**j**), perovskite/Li0.33:CuSCN (**k**), and perovskite/Li0.33:CuSCN/PCPDTBT (**l**) films. White, yellow, and green scale-bars represent 200 nm, 100 nm, and 50 nm, respectively.

**Figure 3 nanomaterials-12-03969-f003:**
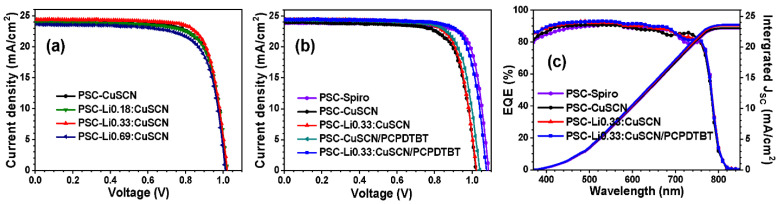
*J-V* curves for PSC devices employing Li:CuSCN HTMs with various Li doping concentrations (**a**). *J-V* curves (**b**) and IPCE spectra (**c**) of PSC devices employing various HTM systems.

**Figure 4 nanomaterials-12-03969-f004:**
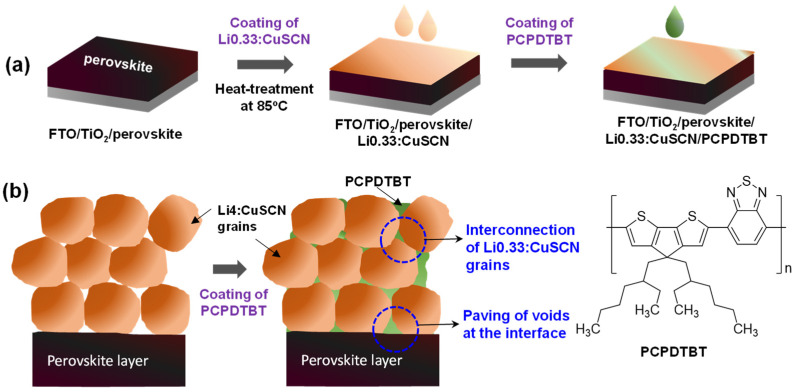
Procedure for PCPDTBT treatment over perovskite/Li0.33:CuSCN (**a**) and a diagram describing the roles of PCPDTBT (**b**).

**Figure 5 nanomaterials-12-03969-f005:**
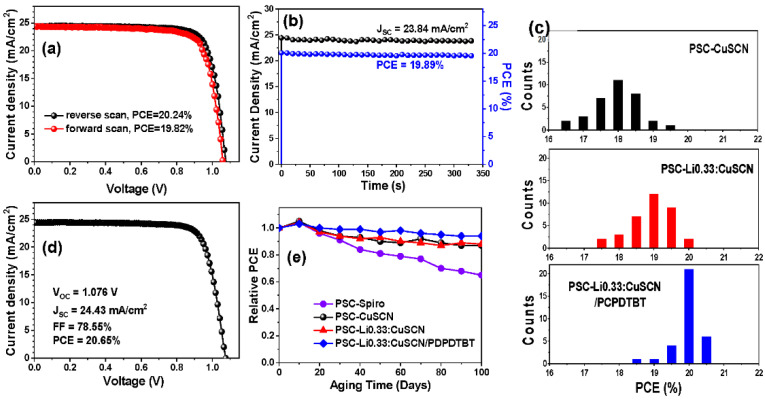
*J-V* curves for forward and backward scans (**a**) and steady-state power curves under continuous illumination, (**b**), of PSC-Li0.33:CuSCN/PDPDTBT devices. (**c**) PCE distributions of PSC-CuSCN, PSC-Li0.33:CuSCN, and PSC-Li0.33:CuSCN/PDPDTBT devices. (**d**) *J-V* curves of champion PSC-Li0.33:CuSCN/PDPDTBT devices. (**e**) Normalized PCEs of PSC-Spiro, PSC-CuSCN, PSC-Li0.33:CuSCN, and PSC-Li0.33:CuSCN/PDPDTBT devices as a function of aging time at 25 °C and 25 ± 5% relative humidity.

**Figure 6 nanomaterials-12-03969-f006:**
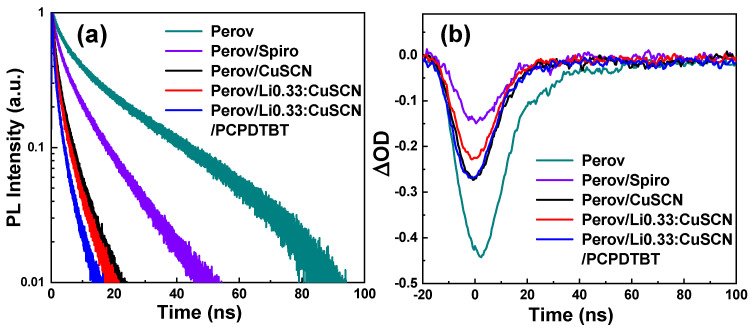
Temporal decays of TR-PL signals (**a**) and TA spectra at 780 nm (**b**) of the bare perovskite (Perov) and various glass/Perov/HTM devices.

**Table 1 nanomaterials-12-03969-t001:** PV parameters of *J-V* curves for the PSC devices employing various HTM systems. The concentration of the PCPDTBT coating solution is 0.2 mg in 1 mL chlorobenzene, if its concentration is not indicated as otherwise.

Kinds of HTMs	*V_OC_* (V)	*J_SC_* (mA/cm^2^)	*FF* (%)	PCE (%)
CuSCN	1.018	23.86	74.24	18.03
Li0.18:CuSCN	1.024	24.01	74.83	18.40
Li0.33:CuSCN	1.032	24.40	75.69	19.06
Li0.69:CuSCN	1.018	23.59	73.55	17.66
Spiro-OMeTAD	1.089	23.93	79.08	20.61
CuSCN/PCPDTBT	1.048	24.28	75.13	19.12
Li0.33:CuSCN/PCPDTBT	1.075	24.41	77.12	20.24
PCPDTBT	0.896	18.21	57.60	9.40
PCPDTBT (20 mg/mL)	0.915	19.35	48.12	8.52
PCPDTBT/Li0.33:CuSCN	1.048	24.34	75.54	19.27

**Table 2 nanomaterials-12-03969-t002:** The TR-PL and TA decay lifetimes of the bare perovskite (Perov) and various Perov/HTM devices. ^a^ The PL lifetimes correspond to the amplitude-weighted average lifetime of a multi-exponential decay fit. ^b^ TA decay lifetimes have been determined by single exponential fitting.

Sample	PL Lifetime (ns) ^a^	TA Decay Lifetime (ns) ^b^
Perov	14.53	14.29
Perov/Spiro	5.43	11.34
Perov/CuSCN	1.90	9.73
Perov/Li0.33:CuSCN	1.45	9.67
Perov/Li0.33:CuSCN/PCPDTBT	0.97	9.75

## Data Availability

Not applicable.

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
