# Peer review of "Formation of Highly Efficient Perovskite Solar Cells by Applying Li-Doped CuSCN Hole Conductor and Interface Treatment"

_nanomaterials, 2022, doi:10.3390/nano12223969_

Round 1

Reviewer 1 Report

Li-Doped CuSCN Hole Conductor and Interface Treatment  in PerSCs was studied. It reports Li can increase the  crystallinity of CuSCN and  the introduction of PCPDTBT onto the  pervskite/Li0.33:CuSCN further improved device PCE and stability. It is import to study the inorganic HTL and this paper get some good results. I suggest it to be published after minor revision.

1. In the experiment section, it was found that 0.15M CuSCN was dissolved in DPS. Howwever, it is too much for CuSCN in DPS as far as I know (~the limitation solubility of  CuSCN in DPS is 6mg/ml, 0.05M, Journal of Materials Science: Materials in Electronics volume 30, pages 11576–11587 (2019)) , how can you get the high solubility. Please make clear in your paper.

2.  Compare fig 2 b and 2 c, it seems that the pinhole in fig 2c is more than that in fig 2 a. It is very important to get the pinhole free HTL for efficent PerSCs becasue the  pinhole results in high recombination. Please make clear this part.

3. the recent process about CuSCN in PerSCs is suggest to be cited, such as 

Journal of Energy Chemistry 62 (2021) 459–476.

Author Response

  1. In the experiment section, it was found that 0.15M CuSCN was dissolved in DPS. However, it is too much for CuSCN in DPS as far as I know [~the limitation solubility of  CuSCN in DPS is 6 mg/mL, 0.05M, Journal of Materials Science: Materials in Electronics, 30, 11576–11587 (2019)], how can you get the high solubility. Please make clear in your paper.

<Answer>

We appreciate thoughtful comment of reviewer. It was our mistake. The solvent used for CuSCN dissolution was diethyl sulfide. We corrected it. Please see line 154 and 157.

  1. Compare Fig 2b and 2c, it seems that the pinhole in Fig 2c is more than that in Fig 2b. It is very important to get the pinhole free HTL for efficient PerSCs because the pinhole results in high recombination. Please make clear this part.

<Answer>

Thank you very much for the important comment.

Fig. 2a shows bare perovskite, and Fig. 2b and Fig. 2c show perovskite/CuSCN and perovskite/Li0.33:CuSCN, respectively. CuSCN has an intrinsic property to form a crystallized structure. Therefore, for the solution processed CuSCN film, texturized CuSCN grains and nanosized pores (or pinholes) are found in general. This is not observed at all for the films prepared by organic HTMs. As the reviewer mentioned, more charge recombination is expected for the CuSCN-applied PSC devices, thereby showing lower VOC and PCE. This is the main reason that CuSCN-applied PSCs reveal poorer PV performance than the PSCs with organic HTMs, although CuSCN has significantly higher hole mobility than organic HTMs.

Furthermore, the Li0.33:CuSCN film exhibits larger grains and higher porosity than the bare CuSCN film, as shown in SEM images. Therefore, its larger grains deteriorate the interconnection among the individual Li0.33:CuSCN grains and the contact with the perovskite surface. As a result, it is likely that more defects are formed at the perovskite/Li0.33:CuSCN interface, leading to increase of charge recombination for the PSC device. In this regard, interface control between perovskite and Li0.33:CuSCN layers and paving of pinholes formed in the Li0.33:CuSCN will be highly important. In this work, by the interfacial treatment with PCPDTBT, we tried to solve this issue and achieved significantly improved PV performance.

In line 273-280, we described as follows.

Although the Li0.33:CuSCN possesses higher hole mobility than the bare CuSCN, the increase of VOC and PCE was not significant. We believe that this is closely related with larger grain size of the Li0.33:CuSCN film formed over the perovskite layer. Its larger grains deteriorate the interconnection among the individual Li0.33:CuSCN grains and the contact with the perovskite surface. As a result, it is likely that more defects are formed at the perovskite/Li0.33:CuSCN interface, leading to increase of charge recombination at the interface. Thus, these two opposite effects cancel each other out, resulting in no significant improvement of VOC and PCE in the PSC-Li0.33:CuSCN device.

  1. The recent process about CuSCN in PerSCs is suggest to be cited, such as  Journal of Energy Chemistry 62 (2021) 459–476.

<Answer>

Thank you for your kind recommendation. We cited recent research works on CuSCN-based perovskite solar cells, as listed below.

  1. Ye, T.; Sun, X.; Zhang, X. Hao, S. Recent Advances of Cu-Based Hole Transport Materials and Their Interface Engineering Concerning Different Processing Methods in Perovskite Solar Cells. J. Energy Chem. 2021, 62, 459–476.
  1. Kim, G.; Kwon, N.; Lee, D.; Kim, M.; Kim, M.; Lee, Y.; Kim, W.; Hyeon, D.; Kim, B.; Jeong, M. S.; Hong, J.; Yang, J. Methylammonium Compensation Effects in MAPbI3 Perovskite Solar Cells for High-Quality Inorganic CuSCN Hole Transport Layers. ACS Appl. Mater. Interfaces 2022, 14, 5203−5210.

Reviewer 2 Report

In the manuscript entitled “Formation of Highly Efficient Perovskite Solar Cells by Applying Li-Doped CuSCN Hole Conductor and Interface Treatment”, the author  provides us with an ideal research object, the PSC device employing 0.33% Li-doped CuSCN (Li0.33:CuSCN) exhibits higher power conversion efficiency (PCE) than the devices with pristine CuSCN and through depositing a slight amount of PCPDTBT over the Li0.33:CuSCN layer, VOC was increased to 1.075 V, resulting in an average PCE of 20.24% and 20.65% for the champion device.This result is meaningful, however, the performance of this work itself is not very outstanding. It was not innovative enough to be published in the Nanomaterials .The reviewer hopes the following issues may do help to improve the manuscript:

1. In this work, It is not meaningful for the author to discuss the increase of open circuit voltage of Li doped devices. The favorable factors leading to the increase in efficiency should be analyzed emphatically.

2. It is pointed out that only a small amount of PCPDTBT remains on the surface of Li0.33: CuSCN film, and a large amount of PCPDTBT exists at the interface between perovskite and Li0.33: CuSCN layer. The corresponding phenomenon cannot be observed by SEM.

3.It is pointed out that the average roughness of Li0.33: CuSCN is not significantly reduced by introducing PCPDTBT, which may be caused by the low concentration of spin-coated PCPDTBT solution.

4.it is more illustrative by Calculating the lag index when discussing the lag, the lag index formula can be listed .

5.In this paper,“It is proposed that the perovskite/CuSCN interface is not significantly changed by PCPDTBT coating, indicating that PCPDTBT has not penetrated the CuSCN layer”.It cannot be determined by SEM, and need some further verification.

6. It is generally believed that Li will have a strong absorption effect on water. The author needs to explain why the stability of the device increases after doping Li.

7. Some typos

(1)In Figure 2, the scale bar was not marked.

(2)In Figure SEM,the diagram was very fuzzy.

(3)In Figure 3,the line weight dimension was too large.

Author Response

  1. In this work, It is not meaningful for the author to discuss the increase of open circuit voltage of Li doped devices. The favorable factors leading to the increase in efficiency should be analyzed emphatically.

<Answer>

Thank you very much for the comment.

As we described in the "Introduction" part, the issue of charge recombination at the interface of CuSCN/perovskite is the most important in determining the PV property of the CuSCN-based PSC (see line 62-72), as many researchers reported  [reference 35, 39, 47-52]. Therefore, open circuit voltage (VOC) is the major factor, because VOC value is directly influenced by charge recombination. However, we acknowledge that other PV parameters are also important.

Hence, in the revised manuscript, we emphasized other PV parameters also.  

The corrections that we made are ss follows.

- In line 13, we removed “but the increase in VOC was not significant.”

- In line 60-61, we changed to “~ significantly lower photovoltaic (PV) performance~”.

- In line 86, we changed to “~ high PV performance~”.

- In line 350, we changed to “~ enhancement of PV performance~”.  

- In line 353, we changed to “~ VOC and PCE ~”.  

  1. It is pointed out that only a small amount of PCPDTBT remains on the surface of Li0.33: CuSCN film, and a large amount of PCPDTBT exists at the interface between perovskite and Li0.33: CuSCN layer. The corresponding phenomenon cannot be observed by SEM.

<Answer>

Thank you very much for the comment. We admit that the decscription of the SEM images is not sufficient to understand. Thus, we described the SEM images in more detail.

In the revised manuscript (line 305-314), we added the following pragraph, and also modified Figure 2h.

Figure 2d, 2g and 2h show plan-view and cross-sectional-view SEM images of PCPDTBT-coated Li0.33:CuSCN (Li0.33:CuSCN/PCPDTBT) films. The plan-view image in Figure 2d shows that the porous structure of Li0.33:CuSCN film is partially paved by PCPDTBT. Also, the cross-sectional image in Figure 2g exhibits that the interface of the perovskite/Li0.33:CuSCN was significantly changed, although PCPDTBT was coated on top of the Li0.33:CuSCN layer. Noticeably, as can be clearly seen in the magnified image (Figure 2h), a new substance is present at the interface. We speculate that the material present in the dashed ellipse in Figure 2h is the PCPDTBT. It is inferred that some of the coated PCPDTBT penetrated the porous Li0.33:CuSCN layer to reach the perovskite/Li0.33:CuSCN interface.

  1. It is pointed out that the average roughness of Li0.33: CuSCN is not significantly reduced by introducing PCPDTBT, which may be caused by the low concentration of spin-coated PCPDTBT solution.

<Answer>

Thank you very much for the comment. In the revised manuscript, we simply described what we observed from the AFM images. Please see line 314-318.

Figure 2j-l shows atomic force microscope (AFM) images of bare perovskite, perovskite/Li0.33:CuSCN, and perovskite/Li0.33:CuSCN/PCPDTBT surfaces. By coating PCPDTBT on perovskite/Li0.33:CuSCN, the average roughness (Rav) of Li0.33:CuSCN was decreased only slightly, suggesting that the amount of PCPDTBT present on the surface of Li0.33:CuSCN layer is not large.

  1. It is more illustrative by Calculating the lag index when discussing the lag, the lag index formula can be listed.

<Answer>

Thank you for the comment. We are sorry, but we do not understand the reviewer’s intent for this comment.

5. In this paper,“It is proposed that the perovskite/CuSCN interface is not significantly changed by PCPDTBT coating, indicating that PCPDTBT has not penetrated the CuSCN layer”. It cannot be determined by SEM, and need some further verification.

<Answer>

Thank you very much for the comment.

We acknowledge that the description of the SEM images was not sufficient. We tried to describe the SEM images in more detail. On the other hand, as another evidence for PCPDTBT penetration, the XPS depth profile results are shown in Fig. S5. Compared to the perovskite/Li0.33:CuSCN film, the perovskite/Li0.33:CuSCN/PCPDTBT film shows a higher carbon content in the region of Li0.33:CuSCN film. The high carbon content indicates the presence of PCPDTBT in the Li0.33:CuSCN film, indicaitng that PCPDTBT penetrated through the Li0.33:CuSCN layer.

In the revised manuscript, we added the following paragraph.

(Line 305-314)

Figure 2d, 2g and 2h show plan-view and cross-sectional-view SEM images of PCPDTBT-coated Li0.33:CuSCN (Li0.33:CuSCN/PCPDTBT) films. The plan-view image in Figure 2d shows that the porous structure of Li0.33:CuSCN film is partially paved by PCPDTBT. Also, the cross-sectional image in Figure 2g exhibits that the interface of the perovskite/Li0.33:CuSCN was significantly changed, although PCPDTBT was coated on top of the Li0.33:CuSCN layer. Noticeably, as seen in the magnified image (Figure 2h), a new substance is present at the interface. We speculate that the material present in the dashed ellipse in Figure 2h is the PCPDTBT. It is inferred that some of the coated PCPDTBT penetrated the porous Li0.33:CuSCN layer to reach the perovskite/Li0.33:CuSCN interface.

(Line 319-329)

XPS depth profiles were obtained to analyze the presence of PCPDTBT throughout the perovskite/Li0.33CuSCN/PCPDTBT multilayered films. As shown in Figure S3, the elemental compositions of Cu, Pb, and C were monitored as a function of etch time. The presence of PCPDTBT can be identified from the distribution of element C. Roughly, the Li0.33CuSCN layer is present in the 0−250 s etch time range and the perovskite layer appears in the etch time range higher than 250 s. The C content in the perovskite/Li0.33CuSCN/PCPDTBT film was significantly higher in the 0−300 s range compared to that in the perovskite/Li0.33CuSCN. This suggests that PCPDTBT is present not only on the top of the Li0.33CuSCN layer, but also inside the Li0.33CuSCN layer and at the interface of the perovskite layer, which is consistent with the observations from the SEM images in Figure 2.

  1. It is generally believed that Li will have a strong absorption effect on water. The author needs to explain why the stability of the device increases after doping Li.

(Answer)

We appreciate very thoughtful comment. Absolutely, we agree with the views of the reviewer. In principle, Li-doping can increase moisture sensitivity.

Indeed, in our experimental results, Li-doping in CuSCN did not increased the stability of the device. Long-term stability of PSC-Li:CuSCN was basically the same as that of PSC-CuSCN. This suggests that a very small amount of Li doping into CuSCN does not affect the device stability. Significant increase of long-term stability in this work was caused by further treatment with PCPDTBT.

Please see  line 401-405.

  1. Some typos

(1)In Figure 2, the scale bar was not marked.

Scale bars are already indicated in the figure caption. Please see the sentence in the last line.

(2)In Figure SEM,the diagram was very fuzzy.

We added “dashed ellipse“ in the SEM image of Figure 2h in order to help reader’s understanding.

(3)In Figure 3,the line weight dimension was too large.

In the revised manuscript, we decreased the size of symbols. Please see Figure 3.

Reviewer 3 Report

The Manuscript entitled, “Formation of Highly Efficient Perovskite Solar Cells by Applying Li-Doped CuSCN Hole Conductor and Interface Treatment” by Yang et.al. is focused on the effects of Li-doping on properties of CuSCN hole transporting material and also the performance of perovskite solar cells. Furthermore, authors showed the effects of penetration of polymeric PCPDTBT into the boundaries of Li:CuSCN grains on overall performance of solar cells. Photoluminescence quenching in Perovskite / Li0.33:CuSCN / PCPDTBT device architecture shows effective charge collection compared to other combinations described in the manuscript. It highlights the role of Li-doping and interface engineering through PCPDTBT polymer. The enhancement in hole mobility of Li-CuSCN also reflects the importance of Li-doping in hole transport materials. 

The results presented in the manuscripts will provides insights to the readers on the effects of elemental doping on properties of HTM and the performance of perovskite solar cells. 

Moreover, the authors need to address the following issues before the publication of this manuscript.

  1. Is there any particular reason for selecting 0.2 mg/mL concentration of PCPDTBT polymer?
  2. Authors discussed the role of polymer in perovskite solar cell devices. Could authors provide an evidence of interaction of polymer with Li-CuSCN?

Author Response

  1. Is there any particular reason for selecting 0.2 mg/mL concentration of PCPDTBT polymer?

(Answer)

Thank you very much for the very thoughtful comment.

In preparing hole-transporting layer by a spin-coating process, a typical concentration of organic HTMs is about 20 mg/mL. In this work, we used a very dilute PCPDTBT in order to avoid hole conduction by PCPDTBT. In fact, pure PCPDTBT exhibits much poorer hole-transporting performance than Li0.33:CuSCN. In preliminary experiments, concentrations of PCPDTBT varied in the range of 0.1 – 1.0 mg/mL, but the effects of PCPDTBT were not significantly different. Hence, 0.2 mg/mL is regarded to be one of the optimized concentrations.

  1. Authors discussed the role of polymer in perovskite solar cell devices. Could authors provide an evidence of interaction of polymer with Li-CuSCN?

(Answer)

Thank you very much for the very important point.

In our previous report, we found that CuSCN and S-terminated organic compound [phenylene-1,4-diisothiocyanate (Ph-DITC)] makes a strong interaction [J. Mater. Chem. A. 2019, 7, 6028−6037.; reference 48 in the revised manuscript]. The FT-IR analysis confirmed evidence for the existence of a significant interaction between Cu in CuSCN and S in Ph-DITC. We also found that the S-terminal group can also strongly interact with the Pb atom of the perovskite. For this reason, we selected PCPDTBT polymers containing numerous S atoms (3 sulfur groups per monomer). We believe that PCPDTBT will have a strong interaction with Li0.33:CuSCN. That is, PCPDTBT will completely pave the pinholes and grain boundaries formed in the Li0.33:CuSCN layer.

In line 330-336, we revised the paragraph as follows.

Consequently, a significant amount of PCPDTBT permeate into the Li0.33:CuSCN grain boundaries (or pinholes), and it induces a tight interconnection of Li0.33:CuSCN grains because sulfur atoms of PCPDTBT can have strong interactions with Cu atom of Li0.33:CuSCN. In addition, PCPDTBT can penetrate down to the interface of the perovskite/Li0.33:CuSCN. Therefore, the non-contacting part at the interface can be paved by the PCPDTBT, resulting in the removal of the defects on the perovskite surface by the passivation of Pb atoms.

Round 2

Reviewer 2 Report

The authors have revised the manuscript according to the questions raised by reviewers, the reviewer will support this work to be published on Nanomaterials.